# A 3K Axiom SNP array from a transcriptome-wide SNP resource sheds new light on the genetic diversity and structure of the iconic subtropical conifer tree *Araucaria angustifolia* (Bert.) Kuntze

**Pedro Italo T. Silva**[1,2¤], **Orzenil B. Silva-Junior**[1], **Lucileide V. Resende**[1], **Valderes A. Sousa**[3], **Ananda V. Aguiar**[3], **Dario Grattapaglia**[1,2,4] *

**1** Plant Genetics Laboratory, EMBRAPA Genetic Resources and Biotechnology, DF, Brasilia, Brazil, **2** University of Brasília, Cell Biology Department, Campus Universitário, DF, Brasília, Brazil, **3** Empresa Brasileira de Pesquisa Agropecuária–EMBRAPA Florestas, PR, Colombo, Brazil, **4** Graduate Program in Genomic Sciences, Universidade Católica de Brasília, Brasília, DF, Brazil

¤ Current address: Corteva Agriscience™ Guarapuava Research Station, PR, Guarapuava, Brazil
* dario.grattapaglia@embrapa.br

**Data Availability Statement:** All relevant data are within the paper and its Supporting Information

## Abstract

High-throughput SNP genotyping has become a precondition to move to higher precision and wider genome coverage genetic analysis of natural and breeding populations of non-model species. We developed a 44,318 annotated SNP catalog for *Araucaria angustifolia*, a grandiose subtropical conifer tree, one of the only two native Brazilian gymnosperms, critically endangered due to its valuable wood and seeds. Following transcriptome assembly and annotation, SNPs were discovered from RNA-seq and pooled RAD-seq data. From the SNP catalog, an Axiom® SNP array with 3,038 validated SNPs was developed and used to provide a comprehensive look at the genetic diversity and structure of 15 populations across the natural range of the species. RNA-seq was a far superior source of SNPs when compared to RAD-seq in terms of conversion rate to polymorphic markers on the array, likely due to the more efficient complexity reduction of the huge conifer genome. By matching microsatellite and SNP data on the same set of *A. angustifolia* individuals, we show that SNPs reflect more precisely the actual genome-wide patterns of genetic diversity and structure, challenging previous microsatellite-based assessments. Moreover, SNPs corroborated the known major north-south genetic cline, but allowed a more accurate attribution to regional versus among-population differentiation, indicating the potential to select ancestry-informative markers. The availability of a public, user-friendly 3K SNP array for *A. angustifolia* and a catalog of 44,318 SNPs predicted to provide ~29,000 informative SNPs across ~20,000 loci across the genome, will allow tackling still unsettled questions on its evolutionary history, toward a more comprehensive picture of the origin, past dynamics and future trend of the species' genetic resources. Additionally, but not less importantly, the SNP array described, unlocks the potential to adopt genomic prediction methods to accelerate the still very timid efforts of systematic tree breeding of *A. angustifolia*.

files. Additionally all raw sequence data are available as follows: 1. All RAD-seq raw sequencing data have been deposited in the NCBI SRA (Short Read Archive) under BioProject, PRJNA602322 at https://www.ncbi.nlm.nih.gov/bioproject/602322 2. Araucaria_snps.RNA-seq.vcf: vcf file containing all SNPs discovered using RNA-seq sequences including those that did not pass the quality filters are available at 10.6084/m9.figshare.11861712 3. Araucaria_snps.RAD-seq.vcf: vcf file containing all SNPs discovered using RAD-seq sequences including those that did not pass the quality filters are available at 10.6084/m9.figshare.11861682 4. Araucaria_RNA-seq_contigs.fasta: fasta archive of contigs from RNA-seq are available at 10.6084/m9.figshare.11861754 5. Araucaria_RAD-seq_contigs.fasta: fasta archive of contigs from RAD-seq are available at 10.6084/m9.figshare.11861718.

**Funding:** This work was supported by competitive grants: "NEXTREE" grant # 193.000.570/2009 by Fundação de Amparo à Pesquisa do Distrito Federal (FAP-DF) (www.fap.df.gov.br) to DG; EMBRAPA (Empresa Brasileira de Pesquisa Agropecuaria) (www.embrapa.br) grant # 02.11.08.005.00.03 to V.A.S and D.G CNPq (Conselho Nacional de Desenvolvimento Científico e Tecnológico) (www.cnpq.br) grants 400663-2012/0 and 308431-2013/8 to DG The funders had no role in study design, data collection and analysis, decision to publish, or preparation of the manuscript.

**Competing interests:** The authors have declared that no competing interests exist.

# Introduction

The development of high throughput genotyping tools based on large numbers of SNP markers (single nucleotide polymorphisms) has become a prerequisite to move to a higher level of precision and genome coverage for the genetic analysis of natural and breeding populations of non-model organisms [1]. Genome-wide genotyping technologies provide exceptional opportunities to advance the understanding of the overall patterns of genetic diversity to drive conservation efforts [2] and inform genomic assisted breeding [3]. Next-generation sequencing (NGS) technologies have facilitated the task of SNP discovery in plant and animal genomes using different approaches that allow genome complexity reduction and more recently methods based on low to ultra-low coverage sequencing of the whole genome. The most traditional and affordable complexity reduction method has been by RNA sequencing (RNA-seq), in which cDNA is made through reverse transcription from only a fraction of the transcribed genome, followed by sequencing and variant calling. Other methods have adopted different approaches of restriction enzyme digestion followed by high throughput sequencing such as Restriction site associated DNA (RAD-seq) [4] the various alternative protocols of genotyping by sequencing (GbS) [5], and targeted enrichment by sequence capture [6]. These methods have allowed not only the discovery of large numbers of SNPs, but also direct SNP genotyping at accessible costs for under resourced plant and animal species [7].

While targeted enrichment by sequence capture do provide generally reliable and portable SNP genotyping data in highly heterozygous species [8], several are the challenges of restriction enzyme based methods for robust SNP genotyping due to variable sequencing coverage, irregular sampling of loci and lack of a reference genome, causing frequent allele dropout and variable genotype reproducibility [9]. The final number of robust and portable SNPs across experiments is typically only a small fraction of the initial set, defeating the alleged cost advantage and possibly biasing genetic diversity measures [10,11]. For high reproducibility, high throughput genotyping fixed content SNP arrays are currently the gold standard and the only validated platform adopted in humans, major animal, crop and forest tree species. With the substantial price reductions of competing technologies [12] and the possibility of designing multi-species SNP arrays [13], these platforms have become accessible at a fraction of what the cost used to be, translating to equivalent or lower price per informative data point when compared to GbS methods [11].

*Araucaria angustifolia* (Bertol.) Kuntze is an iconic long-lived subtropical conifer tree endemic to Southern and Southeastern Brazil and to minor areas in Argentina and Paraguay. It stands out as the keystone gymnosperm species native to Brazil and the leading species in the mixed Ombrophylous Forest (a.k.a. Araucaria Forest) [14]. Currently, with a strong reduction of its original old-growth forest area, the Araucaria Forest biome is one of the most threatened in Brazil, with *A. angustifolia* also called Paraná Pine included as critically endangered in the IUCN Red List of Threatened Species [15]. Besides its ecologically keystone role, *A. angustifolia* had a historically important social and economic role during the European colonization of Southern Brazil [16] and as a outstanding looking tree it is frequently planted for its ornamental appearance in gardens and homes for its aesthetic value. Isozymes allowed the first estimates of genetic diversity, structure and mating system [17–21]. Next, studies were carried out using dominant AFLP markers [22–24], or small sets of five to 15 microsatellites to compare the genetic diversity among natural and planted forest stands or estimate spatial genetic structure, mating system and gene flow [22,25–31]. The ultimate goal of these studies has been to provide evidence-based information for supporting conservation strategies. Generally, high levels of genetic diversity have been found, suggesting that *A. angustifolia* is resilient to forest fragmentation, maintaining adequate diversity for sustainable evolution [30,32].

Notwithstanding the existing information on the population genetics of *A. angustifolia*, marker resources and data gathered thereof are still restricted to very few microsatellite loci. Due to their small number, ascertainment bias and mutational behavior, microsatellites are limited as reliable predictors of genetic variation and historical demography of natural populations [33–35]. Clearly, the current molecular toolbox for the analysis of sequence variation in *A. angustifolia* is insufficient for answering important remaining questions, or else, corroborating or challenging the current view on the levels, distribution and dynamics of the current genetic diversity in this iconic subtropical conifer.

In conifers, due to relatively high past implementation costs of fixed content SNP arrays, moderate to high density chips with hundreds or up to several thousand markers have been developed exclusively for the main stream commercially relevant genera for which funding is abundant. These have included *Picea* spp. [36–39], *Pinus* spp. [38,40–43], *Cunninghamia* spp. [44], *Cryptomeria japonica* [45,46] and *Pseudotsuga menziesii* [47]. More recently, however, with advances in technology and cost reductions, SNP arrays have become very competitive with alternative sequence based SNP genotyping methods. In this study, we describe the development of a large annotated SNP catalog for *A. angustifolia*, together with an Axiom® SNP array with ~3,000 validated SNPs. The array was subsequently used to provide a comprehensive look at the genetic diversity and structure of population samples across the entire natural geographical range of the species in Brazil and compare the estimates with microsatellite markers genotyped on the same individuals. The Axiom SNP array described is fully public and accessible to anyone interested.

## Materials and methods

### Plant material and RAD-seq data

RAD-seq libraries were prepared from two genomic DNA samples. Taking advantage of the haploid biology of conifers and the large *A. angustifolia* seeds, the first DNA sample was extracted from a single haploid megagametophyte. The second sample was an equimolar pool of DNA extracted from diploid needle tissue of twelve unrelated trees to serve as a representative sample of diversity for SNP detection. Genomic DNA was extracted with an optimized protocol for challenging samples [48]. The two genomic DNA samples were sent to Floragenex (Portland, OR, USA) for RAD-seq [4], with the difference that the restriction enzyme *PstI* was used for plant genome complexity reduction. The sheared, sequencer-ready fragments were size selected (~200–500 bp) and the RAD-seq libraries sequenced on a Genome Analyzer II (Illumina, San Diego, CA), paired-end 2 x 100 bp mode. RAD tags were used for SNP discovery and selection of SNPs. SNPs were evaluated by a population survey involving a set of 185 35-year old trees sampled in a provenance-progeny trial established in an experimental station in Itapeva, São Paulo State (23°58′56″S 48°52′32″W) with seeds collected in 15 different natural populations covering the geographical range of the species (Fig 1 and S1 Table). Sample collection from natural populations for this genetic study (authorization number 02001.007609/2012-77) was issued by the Brazilian Institute of the Environment (IBAMA), the regulating body of the Brazilian Ministry of Environment. Genomic DNA for downstream SNP and microsatellite genotyping was extracted from needle or bark tissue with the same protocol as for the RAD-seq samples.

### RNA sequence data

RNA-seq sequence data with insert size of approximately 200 bp was generated by Elbl et al. [49] and obtained from the NCBI SRA (short read archive) repository. The plant tissue material used for cDNA sequence data generation consisted of three pools of seeds from three

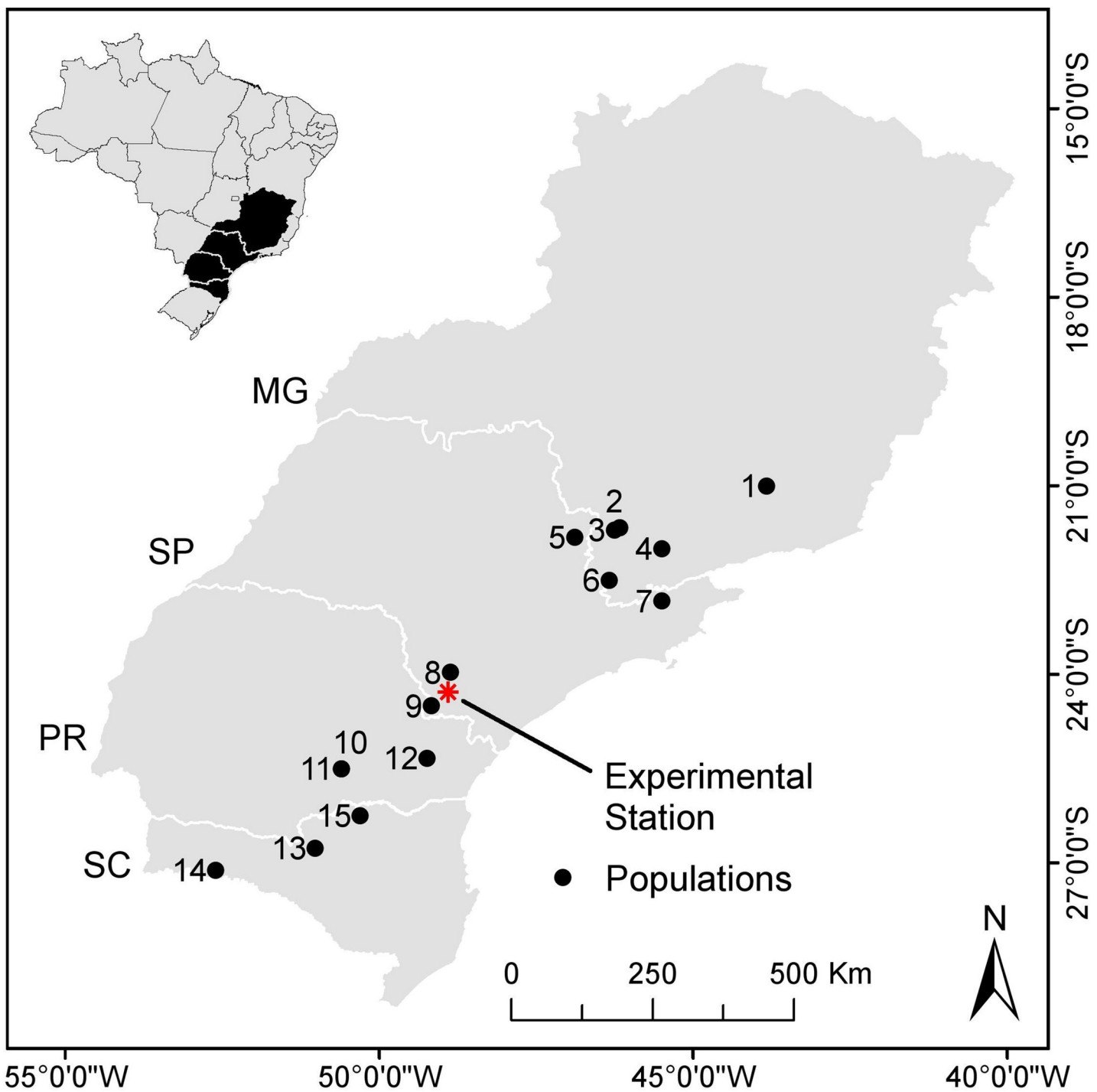

**Fig 1. Geographic distribution of the 15 *Araucaria angustifolia* populations studied.** Population code used and respective locations are: 1.BAR: Barbacena–MG; 2. IPI: Ipiúna de Calda, MG; 3.CON: Congonhal, MG; 4. LAM: Lambarí, MG; 5.VAR: Vargem Grande do Sul, SP; 6.CAM: Camanducaia, MG; 7.CJO: Campos do Jordão, SP; 8.ITA: Itapeva, SP; 9.ITR: Itararé, SP; 10.IRA: Iratí, PR; 11. IRT: Iratí (Tardio), PR; 12.QBA: Quatro Barras, PR; 13.CAC: Caçador, SC; 14.CHA: Chapecó, SC; 15: TRB: Três Barras, SC. Indicated also the experimental station where the provenance/progeny field trial was established and actual samples collected for the population survey.

different megastrobiles and three somatic embryogenic cultures of *A. angustifolia* as described. The raw dataset of 642 million 100 bp reads was downloaded, quality filtered down to 326 million reads, and used for *de novo* transcript assembly and SNPs discovery. Due to the large size (1C = 22 Gb) and highly repetitive nature of the *A. angustifolia* genome [50], a thorough analytical procedure for SNP discovery and ascertainment was adopted to maximize the likelihood of successful downstream SNPs genotyping.

## Transcriptome assembly

As a first step we used the StringTie [51] method to reconstruct cDNA fragments for both end of the reads to pre-assemble longer contigs from the RNA-Seq paired-end data aiming to achieve the better performance of the subsequent transcriptome assembly. From StringTie +SR, we ran the perl script SR (http://ccb.jhu.edu/software/stringtie/dl/superreads.pl) separately on the reads of each one of the three megastrobiles and three somatic embryogenic cultures of *A. angustifolia*. The SR method, a super-reads module borrowed from MaSuRCA [52], pre-assembled *de novo* the paired-end reads from the unambiguous, non-branching parts of the transcript in these six sources of cDNA data, resulting in super-reads of approximately 300 bp, corresponding to the entire sequence of the original cDNA fragment. These super-reads were used as input of the miraSearchESTSNPs from MIRA 4 assembler [53] to generate the consensus sequences of the transcript assembly from the different sources of cDNA and to perform the SNP detection within this assembly. Afterwards, miraSearchESTSNPs from MIRA 4 assembler was used to generate a transcript assembly from the super-reads. The assembly was inspected for any positions with conflicts that could not be resolved automatically by MIRA These suspicious positions were collected and formatted as a single file containing coordinates along the contigs in BED format. The consensus sequences in the assembly were further evaluated with TransRate [54] to detect contigs that exhibited signs of chimerism, structural errors, incomplete assembly and transcripts that were likely paralogs.

## RAD sequences alignment and variants discovery

RAD-seq data was analyzed using a reference-free bioinformatics strategy and parameters described in Senn et al. [55] to end with sufficient flanking region around the SNPs to allow adequate probe design for the fixed content SNP array. To make use of all the sequence information when building the catalogue, the RAD sequenced samples were treated as two 'parent' individuals, named *single_haploid* and *pool_diploid*, respectively. Based on the catalog of formed loci, Read 2 tags were collated separately for each 'parent'. To avoid SNP calls at low coverage for Read 2 tags, for downstream analyses we only kept loci where 20 or more reads were available in total. Afterwards, we used CORTEX_VAR [56] to simultaneously form the entire sequence of loci and calling of variants using Read 2 tags that had representation across the two 'parent' individuals. Again, we followed standard steps described in Senn et al. [55] to simultaneously assemble contigs for each of the loci from collated Read 2 tags, to identify putative SNPs, to characterize the actual alleles of both 'parents' individuals and to genotype. We used the output of CORTEX_VAR to form a pseudo-reference genome containing one 'chromosome' per identified SNP, whose sequence provided the 5-prime flank, the SNP alternative alleles and the 3-prime flank for the variant in the 'parent' named single. We then processed the output of CORTEX_VAR to build a VCF-like file using this pseudo-reference to provide a coordinate system using the script process_calls.pl in CORTEX_VAR. This script collects the information on the variant calls, alleles and their read coverage and the most likely genotype at each SNP/ InDEL for each 'parent' individual, along with a genotype confidence (GT_CONF) using a maximum likelihood approach against the confidence of the second most likely genotype.

## RNA-seq alignment and sequence variants discovery

Contigs in the transcriptome assembly were used as reference to align back the original RNA-seq paired-end reads. We used only well represented contigs in the sequence reads in the Transrate analysis. We used the Novocraft Version 3.03 (Novocraft Technologies Sdn. Bhd., Malaysia) [57] suite of programs to perform all the alignments to the reference assembly, using standard paired-end processing with base quality calibration options disabled and setting SAM ouput format using "-o SAM". SAM formatted files were subsequently converted to BAM file of alignments for the reads from the eight cDNA libraries. To make use of all sequence information for polymorphism screening, samples were treated as coming from two 'parent' individuals, named *single_haploid* and single_*diploid* respectively. While all seeds were collected from a single individual tree, being the tree outcrossed and thus highly heterozygous, the haploid samples embodied within-tree variation and the diploid both the within and between tree variation. A single BAM file of alignments with read group information for the two contributing samples was processed for marking of duplicates and sorted using Picard [58] and Indel Realignment. The output BAM file was then processed for SNP calls and genotyping using GATK HaplotypeCaller analysis [59]. A VCF file of putative variants (SNPs/InDels) was ultimately obtained.

## SNP filtering and prioritization for array design

VCF files from the sequence variant discovery carried out on RAD-seq and RNA-seq data were further refined to retain only positions in the references denoting putative high-quality SNPs. First we inspected each variant and set the genotype calls to NULL for samples where the genotype confidence (GT_CONF) was lower than three and genotype quality (GQ) lower than 30 for the RAD-seq and RNA-seq analyses, respectively. Variant sites without any genotype call across the two samples were removed from the VCF files. Then, several filter tags were applied to the remaining putative variant sites by inspecting their sequence vicinity and context. Each variant classified as InDel was tagged as *TypeIndel*. Each SNP variant located in the vicinity, 60-bp in each direction, of an InDel was tagged as *SnpGap*. Additionally, if more than one SNP variant was found within a 30 bp window on both sides of the target variant, all of them were deemed hotspots of clustered SNPs and tagged as *snpCluster*. Additionally, as low complexity regions in the DNA sequences contribute to highly variable variant calls between callers [60], we inspected the set of reference sequences targeting the identification of low complexity regions (LCRs) and a BED file of LCRs and repeats was generated and all the SNPs within their coordinates were tagged as *LowComplexityRegion*. For SNPs in the RNA-seq analysis, GATK's suggested hard filter criteria were applied so that variants that failed the filter were tagged as *FailureOnGatkHardFilter*. Taking advantage of the haploid biology of *Araucaria*, we set a tag named *FailHaploidTest* at each variant site where the haploid sample genotype was declared as heterozygous opposed to homozygous (when a genotype was emitted) or NULL (in the case of absence of a confident genotype). Worth to mention that the same SNP could end up having several of these tags since the filtering steps were simultaneous. Finally, we only used A/C, A/G, C/T and G/T polymorphisms as these require a single probe therefore optimizing the space available on the array. To adhere to the Axiom's platform recommendation for probe design, we extracted 35-bp sequences in each direction of any target SNP variant that passed all the filtering tags. All the 71-bp long sequences that likely encompassed a single putative SNP variant were inspected against the complete set of reference sequences obtained by the concatenation of contigs from the transcriptome assembly and pseudo-reference from the RAD-seq loci. Probes were then classified as not recommended when more than 100 matches of any 16-mers in its sequence were found in the reference.

## Functional annotation and classification of the SNP catalog

The full collection of 22,983 sequences representing RAD loci and RNA transcripts from RAD-Seq and RNA-Seq assemblies, respectively, containing SNP variants with the status of "PASS" was subjected to further characterization. First, the sequences were corrected for possible frameshifts and missing signals using default parameters in FrameDP [61] and then characterized for protein similarity & classification, structural properties, gene ontology and annotation using Blast2GO 5 PRO [62]. We used Blast2GO to run BlastX against the Non-redundant protein sequences (nr), with an e-value cutoff of 1.0E-3, word size of three, a maximum number of hits of 20 and using the application low complexity filter. InterProScan was also ran to search for protein families, domains and sites against several databases–CDD, HAMAP, HMMPanther, HMMPfam, HMMPIR, FPrintScan, BlastProDom, ProfileScan, HMMTigr, Gene3D, SFLD, SuperFamily and MobiDBLite. A GO mapping and annotation of the protein Blast hits was carried out against curated Gene Ontology annotated proteins using default parameters.

## SNP array design and validation

For the 44,318 assayable SNPs potential probes were designed for each one in both the forward and reverse direction. A set of 3,400 randomly selected SNPs among the 44,318 was submitted to ThermoFisher to populate the fixed content SNP array. This number of probes was defined based on the available space on a multispecies Axiom® myDesign ™ array that contained a total of 51,867 SNPs, shared among five different plant species, significantly reducing the individual sample genotyping cost while at the same time allowing access to a high quality SNP array for the underfunded species *A. angustifolia* [63]. The four additional species on the array besides *Araucaria angustifolia* (order Pinales) not only belong to different families but also to different phylogenetic orders to minimize the possibility of genome sequence homology. These were: *Anacardium occidentale* (cashew–order Sapindales), *Manihot esculenta* (cassava–order Malpighiales), *Coffea robusta* (coffee–order Gentianales) and *Eucalyptus* sp. (order Myrtales). Additionally, all probes designed for the five species were subject to a detailed sequence evaluation to avoid SNP probe cross talking.

SNP genotyping of a total of 192 samples, 185 unique and seven duplicated for reproducibility estimates, was carried out at ThermoFisher (Santa Clara, CA) and data analyzed using the Axiom Analysis Suite 3.1 [64]. Samples with a dish quality control (DQC) value >0.82 and call rate, CR >0.97 following the recommended "Best Practices Workflow" were considered to have passed the sample quality control assessment. Two criteria were used to classify successfully genotyped SNPs and converted polymorphic SNPs. The first stricter criteria set, suggested as default by the Axiom Analysis Suite used in human SNP evaluation, required a SNP CR > 97%. By this method, SNPs were classified into the following categories: (i) PHR (Polymorphic High Resolution) when the SNP passes the quality criteria and polymorphism measured by the presence of the minor allele in two or more samples, which, for N = 185 translated to a minimum allele frequency MAF≥ 0.005; (ii) MHR (Monomorphic High Resolution) when the SNP passes the quality criteria except for polymorphism; (iii) CRBT (Call Rate Below Threshold) when the SNP CR was <97%; (iv) NMH (No Minor Homozygote) when the SNP passes all QC but only two genotype clusters are detected, (v) OTV (Off-Target Variant) where additional clusters arise from unaccounted sequence variants in the SNP flanking region, and (vi) OTH (Other) when the SNP can't be classified into any of the previous categories. Under this strict classification, only SNPs in categories PHR, MHR and NMH are retained as successful SNPs, while the SNPs in PHR class are those considered converted. The second criteria to declare a successful SNP adopted a more liberal CR >90% commonly used

for plant and animal SNP genotyping. The same MAF$\geq$ 0.005 as in the Axiom criteria was used to declare a polymorphic SNP.

### Genetic diversity and structure analyses with microsatellites and SNPs

To assess the performance of the SNP array for population genetics analyses, the same 185 trees were also genotyped with a set of 8 previously published and widely used *A. angustifolia* microsatellites [65]. Microsatellite genotyping was carried out by fluorescence detection using previously described protocols [66]. Briefly, PCRs were carried out in tetraplex systems with primers labeled with fluorochromes (6-FAM, NED, VIC) and the PCR mixture with ROX-labeled size standard [67] electroinjected in an ABI 3100XL genetic analyzer and data collected using GeneMapper (Thermo Fisher Scientific). SNP and microsatellites estimates of allele frequencies, population diversity, i.e. observed ($H_o$) and expected ($H_e$) heterozygosity, coefficient of inbreeding ($F_{is}$) based on Weir and Cockerham inbreeding estimator (*f*) [68] and the fixation index as a measure of population differentiation ($F_{st}$) with their respective 95% confidence intervals were obtained using GDA (Genetic Data Analysis) [69]. AMOVA (Analysis of Molecular Variance) and Principal coordinate analyses (PCoA) plots were obtained using GENA-LEX 6.501 [70]. Genetic structure was further assessed using the approach implemented by STRUCTURE [71] under an admixture model with correlated allele frequencies. Jobs were run applying a burn-in length of 50,000 and 50,000 iterations for data collection with K ranging from 2 up to 15 potentially inferred clusters, performed with 10 independent runs each. For the microsatellite data and reduced 80 SNPs set the analysis was carried out using STRUC-TURE v2.3.4 on a single computer while for the large SNP data set a multi-core computer was used with ParallelStructure [72]. Outputs of STRUCTURE were used to define the most probable number of K clusters by the 'ΔK' metric [73] using Structure Harvester [74] and to generate a consensus solution and plots of the 10 independent runs by a Markov clustering algorithm implemented by CLUMPAK [75].

## Results and discussion

### RAD-Seq vs RNA-seq for SNP discovery in a complex genome

Using RAD-seq, 44,332,020 and 15,920,336 reads were generated from the single sample hap-loid library and the multi-sample pooled library, respectively, totaling 60,252,356 reads. Reads from the single sample haploid library were used for the assembly of a haploid pseudorefer-ence. From all 60,252,356 reads, a total of 176,629 contigs with size >120bp and average size of 325 bp was obtained and used for SNP discovery. For the RNA-seq resource, from the full set of 642 million RNA-seq reads, after applying the quality filters, a total of 326 million reads were ultimately used in transcriptome assembly, resulting in 43,608 unique contigs. The total number of raw sequence variants discovered following alignment and sequence variants dis-covery were 17,428 for RAD-seq and 309,509 for RNA-seq. The simultaneously applied quality filter steps adopted excluded the vast majority of SNPs due to failure in one or more criteria adopted for SNP selection (Table 1).

As expected, the genome complexity reduction generated by RAD-seq provided a much larger number of contigs on a per single quality read basis (176,629/60,252,356 = 0.29%) than the RNA-seq source (43,608/326,000,000 = 0.013%). On the other hand, RNA-seq was three times more efficient in providing raw sequence variants (309,509/326,000,000 = 0.095%) than RAD-seq (17,428/60,252,356 = 0.029%), despite the fact that a smaller number of individual trees were represented in the sequenced sample. This can partly be explained by the fact that the length and sequence coverage of each RAD sequence contig is shorter that RNA-seq such that the final SNPs count is lower, and only few eventually abide to the flanking sequence

**Table 1. Summary of the number of single-nucleotide polymorphisms (SNPs) filtered out from each sequence source (RAD and RNA sequences) following the simultaneous filters applied for SNP selection toward the construction of the *Araucaria angustifolia* 3K SNP Axiom® Array.**

| | RAD-seq | RNA-seq |
|---|---|---|
| Number of high quality reads used for contig assembly | 20,720,596 | 326,000,000 |
| Number of high quality reads used SNP discovery | 60,252,356 | 326,000,000 |
| Number of contigs used for SNP detection | 176,629 | 43,608 |
| Total number of raw SNPs discovered | 17,428 | 309,509 |
| **Simultaneously applied refining filter for SNP exclusion** | | |
| Number of SNPs located in contigs matching known transposon | - | 40,538 |
| Number of SNPs failing GATK best practice filter | - | 45,279 |
| Number of SNPs failing haploid test (heterozygous in haploid sample) | 5,226 | - |
| Number of SNPs located in homopolymer regions | 763 | - |
| Non assayable InDel variants | 580 | 39,204 |
| Number of SNPs with < 35 pb of available flanking sequence for probe design | 1,818 | 984 |
| Number of SNPs located in low complexity region | 162 | 3,862 |
| Number of SNPs failing genotype confidence score | 3,217 | 0 |
| Number of SNPs in sequences with similarity to other sequences | 0 | 9,332 |
| Number of SNPs with additional SNPs within a 30 bp window on both sides | 3,916 | 236,723 |
| Number of SNPs located up to 60 bp from an indel variant | 49 | 75,374 |
| Number of SNPs located in suspicious contigs according to TransRate evaluation | - | 1,538 |
| **Total Number of retained SNPs** | **4,508** | **39,810** |

requirements of a SNP array. When all filters are considered, the final efficiency of RNA-seq in terms of assayable SNPs per quality raw sequence read was 1.6X higher than RAD-seq (0.012% versus 0.0075%). RNA-seq or exome-capture with RNA-derived probes has been the method of choice for SNP discovery and SNP array development in the very large and complex conifer genomes [38,39,42,43,76]. Our results confirm that RNA-seq is an efficient strategy for the identification of bona fide sequence variants. Additionally, our results also show that RNA-seq was far superior to RAD-seq in terms of coverting these putative variants to polymorphic SNPs, at least under the discovery and SNPs selection criteria used here.

## Functional annotation of the SNP containing sequences

The full catalog of SNP variants retained following all quality filters contained 4,508 and 39,810 positions from RAD-Seq and RNA-Seq data respectively (Table 1). This total set of 44,318 assayable SNPs were located in 4,508 and 18,475 unique reference loci and contigs in the RAD-Seq and RNA-Seq *de novo* assemblies, respectively (S1 File). Therefore out of the 43,608 RNA-seq contig, only 18,475 (42%) could be actually sampled as far as identifying SNPs that passed all the quality and requirement filters. However, these 18,475 RNA-seq contigs provided more than one assayable SNP with an average of 2.15 SNPs and up to 19 SNPs in a single contig, thus providing a rich source of well curated SNPs for genotyping a reasonable portion of the *A. angustifolia* gene space. The average RNA-seq contig size was 1,729 bp, varying between 114 and 12,890 bp. This consolidated resource of 22,983 unique sequences was characterized for protein similarity, structural properties and gene ontology. Out of the 22,983 sequences, 15,144 (66%) had similarity at the used e-value threshold to database proteins. We also found that 9,316 sequences (41%) had InterProScan hits, from which, 5,456 had corresponding GO terms. The gene set covers 1,945 identified domains across 3,231 InterPro protein families, the largest of which being the family P-loop containing nucleoside triphosphate hydrolase with 365 sequences (S1 Fig). According to the similarity reported for the top hits in

the BlastX searches against the nr database, we found that 10,889 sequences (47%) had significant similarity with average of 78% (e-value less than 1e-45) with 10,183 sequences with score greater than 60%. From a species perspective, we found that the highest proportion of top hits against the nr database matched to the gymnosperm *Picea sitchensis* (5,301; 34%), followed by matches to the basal angiosperm *Amborella trichopoda* (1,495; 10%). We assigned a total of 54,297 gene ontology terms to 13,351 (58%) of the putative transcripts of protein-coding genes. Most of the assignments (22,219; 41%) belonged to the 'Cellular Component' category, while the remaining belonged to the 'Biological Process' (19,472; 36%) and the 'Molecular Function' categories (12,606; 23%) (S2 Fig and S1 File). In addition to the markers developed and validated, the newly established transcriptome for *A. araucaria* is similar to other published conifer transcriptomes in terms of size (54 Mbp), number of contigs (43,608 transcripts) and average transcript length (1,205 bp) [77]. Additionally, the score given by the TransRate to the assembly (score > 0.3) indicates further evidence of the assembly quality [54].

## SNP genotyping performance on the Axiom array

From the full catalog of 44,318 putatively assayable SNPs, 3,400 SNPs (2,565 from RNA-seq and 835 from RAD-seq) were randomly selected at a rate of one SNP per unique contig to maximize transcriptome coverage, and in a few cases up to two SNPs for longer contigs (S1 File). Probes for these 3,400 SNPs were evaluated by the array manufacturer (Thermo Fisher) based on the company's *in silico* scoring system that uses a proprietary software that calculates a 'p-convert' value for each submitted SNP, i.e. the probability of a given SNP converting to a reliable SNP assay. Of the 3,400 SNP probes, 3,224 were classified as recommended, 170 as neutral and only six not recommended. In other words, 99.8% of the designed SNP probes successfully abided to the parameters of the Axiom array technology. The 3,400 probes were deliberately randomly selected from the entire 44,318 probes set such that the *in silico* evaluation would provide a bona fide estimate for the predicted success of all developed SNP probes. Assuming this same success rate of 99.8% for the entire SNP catalog, a larger SNP array could be designed to include at least ~40,000 SNPs covering all 22,983 transcriptome contigs.

Out of the 3,400 SNPs tested, using a global success based on a CR ≥ 90%, 3,038 SNPs (89.4%) were successfully genotyped (Table 2 and S2 File) with an average reproducibility rate of 99.95%, ultimately constituting the operational 3K SNP array. Using a global rate based on the Axiom criteria (the sum of PHR, MHR and NMH SNPs) 2,521 SNPs were successfully

**Table 2. Summary of performance of SNPs derived from different sources (RAD-seq and RNA-seq) according to the two different performance criteria adopted (see Methods for details).**

| | RAD-seq | % | RNA-seq | % | Total | % |
|---|---|---|---|---|---|---|
| Total number of SNPs assayed | 835 | | 2,565 | | 3,400 | |
| **Conventional performance criteria** | | | | | | |
| Global success rate—SNPs with Call Rate ≥ 90% | 731 | 87.5 | 2,307 | 89.9 | 3,038 | 89.4 |
| Conversion rate—SNPs with Call Rate ≥ 90% and MAF≥ 0.005 | 62 | 7.4 | 1,960 | 76.4 | 2,022 | 59.5 |
| **Axiom performance criteria** | | | | | | |
| Global success rate—Sum of PHR, MHR and NMH | 626 | 75.0 | 1895 | 73.9 | 2521 | 74.1 |
| PolyHighResolution (PHR) | 4 | 0.5 | 1,646 | 64.2 | 1,650 | 48.5 |
| Mono High Resolution (MHR) | 607 | 72.7 | 207 | 8.1 | 814 | 23.9 |
| No Minor Homozygous (NMH) | 15 | 1.8 | 42 | 1.6 | 57 | 1.7 |
| Call rate below threshold | 13 | 1.6 | 165 | 6.4 | 178 | 5.2 |
| OTV | 28 | 3.4 | 5 | 0.2 | 33 | 1.0 |
| Other | 168 | 20.1 | 500 | 19.5 | 668 | 19.6 |

genotyped (74.1%). When considering a conversion rate (CR $\geq$ 90% and MAF$\geq$ 0.005), 2,022 SNPs (59.5%) were converted and subsequently used in population genetic analyses. With the Axiom quality criteria, only the 1,650 PHR SNPs (48.5%) would be considered converted. RNA-seq was a significantly more efficient source of polymorphic SNPs (76.4%) compared to RAD-seq (7.4%). Due to the size, repetitive nature and lack of quality genome assembly with high continuity/contiguity, success rates of SNP genotyping for conifers in fixed content arrays have been generally slightly lower than those reported for other plant or animal species, varying between 60% and 85% [39,40,43]. Overall, our rates of 73.9% to 76.4% conversion for RNA-seq SNPs are in the similar range of those reported for other conifers SNP arrays derived from RNA-seq data. More importantly, if one applies the global and conversion rates obtained from RNA-seq data to all 39,810 RNA-seq derived SNPs in the catalog, we predict that this resource should provide ~29,000 polymorphic SNPs with CR $\geq$ 90% and MAF$\geq$ 0.005, or ~25,000 PHR converted SNPs.

## SNP polymorphism across populations

From the analysis of a sample of 185 individuals, the site frequency spectrum (SFS) of all successfully genotyped SNPs showed enrichment toward higher frequency SNPs (S3 Fig), possibly because of the criteria used to maximize genotyping success. The rare SNP category, with MAF $\leq$ 0.005 shows that the array also contains a large number of rare SNPs, most of them coming from the RAD-seq data discovery. Despite the fact that typically only SNPs with MAF $\geq$ 0.05 are contemplated, we deliberately intended to take into account rare SNPs in our array for two reasons: (1) rare SNPs could be informative for some specific applications such as gene flow, parentage and association studies; (2) we used stringent dish quality and call rates metrics such that the quality of the genotype calls are robust, despite the low MAF; and (3) by having genotyped limited samples of individuals for each one of several populations in this first study, and in light of the strong differentiation detected (see below), it is expected that SNP allele frequencies will vary across populations, such that a SNP allele may be rare overall but it may have higher frequency and thus be informative in specific populations. So, these rare SNP alleles could turn out to be well represented in some populations when larger samples sizes are genotyped, especially for widespread outbred forest tree populations across large geographical areas.

Just like the commonly adopted practice of selecting the most polymorphic microsatellite markers for routine use leads to an ascertainment bias [78], SNP arrays also experience this trend caused by the SNP discovery and selection process. It has been shown, however, that multiple ways exist to correct ascertainment bias [79], and that typically it affects more profoundly analyses involving scans of selection signatures while population differentiation and diversity analysis are relatively robust to it [80]. Looking at the distribution of MAF for all 2,022 polymorphic SNPs in the samples analyzed (S3 File), a minimum of 1,415 in Population LAM and a maximum of 1,913 in population CHA are polymorphic, with an average of 1,703 per population. These results suggest that no relevant ascertainment bias toward any population exists in the array such that it should provide high-resolution power for detailed intrapopulation genetic analyses (e.g. mating system and kinship) across the natural range of the species.

## Population genetic diversity: SNPs versus microsatellites

Allele frequencies, heterozygosities and inbreeding coefficient by marker in each population are provided as supplementary files for SNPs (S3 File) and microsatellites (S4 File). Individual population and overall heterozygosities, both observed and expected, were more than twice

**Table 3. Comparative summary of genetic diversity parameters ($H_o$ observed heterozigosity; $H_e$ expected heterozygosity) and inbreeding coefficient ($F_{is}$) with its respective 95% confidence interval (C.I) obtained with GDA for the two different data sets for the 15 *A. angustifolia* populations.**

| # | Region[a] | Population | Microsatellite data analysis (8 loci) | | | | | | SNP data analysis (2,022 loci) | | | | | |
|---|---|---|---|---|---|---|---|---|---|---|---|---|---|---|
| | | | $H_o$ | $H_e$ | $F_{is}$# | Lower 95% C.I. | Upper 95% C.I. | Sign.# | $H_o$ | $H_e$ | $F_{is}$# | Lower 95% C.I. | Upper 95% C.I. | Sign.# |
| 1 | N | BAR | 0.557 | 0.763 | 0.276 | 0.100 | 0.495 | * | 0.234 | 0.241 | 0.029 | 0.013 | 0.046 | * |
| 2 | N | IPI | 0.601 | 0.695 | 0.140 | 0.000 | 0.312 | ns | 0.272 | 0.275 | 0.011 | -0.005 | 0.027 | ns |
| 3 | N | CON | 0.594 | 0.660 | 0.105 | -0.128 | 0.307 | ns | 0.270 | 0.271 | 0.000 | -0.018 | 0.020 | ns |
| 4 | N | LAM | 0.615 | 0.656 | 0.066 | -0.026 | 0.162 | ns | 0.261 | 0.252 | -0.037 | -0.020 | -0.056 | * |
| 5 | N | VAR | 0.509 | 0.585 | 0.135 | -0.161 | 0.522 | ns | 0.276 | 0.261 | -0.059 | -0.044 | -0.076 | * |
| 6 | N | CAM | 0.673 | 0.672 | -0.001 | -0.128 | 0.113 | ns | 0.257 | 0.254 | -0.010 | -0.029 | 0.007 | ns |
| 7 | N | CJO | 0.558 | 0.664 | 0.166 | -0.001 | 0.399 | ns | 0.268 | 0.310 | 0.140 | 0.123 | 0.157 | * |
| 8 | S | ITA | 0.719 | 0.765 | 0.064 | -0.068 | 0.181 | ns | 0.354 | 0.337 | -0.054 | -0.037 | -0.071 | * |
| 9 | S | ITR | 0.767 | 0.752 | -0.020 | -0.122 | 0.082 | ns | 0.354 | 0.354 | -0.001 | -0.015 | 0.014 | ns |
| 10 | S | IRA | 0.700 | 0.742 | 0.060 | -0.006 | 0.118 | ns | 0.355 | 0.353 | -0.004 | -0.019 | 0.011 | ns |
| 11 | S | IRT | 0.655 | 0.731 | 0.108 | -0.090 | 0.296 | ns | 0.358 | 0.357 | -0.004 | -0.018 | 0.009 | ns |
| 12 | S | QBA | 0.756 | 0.653 | -0.172 | -0.103 | -0.255 | * | 0.350 | 0.348 | -0.006 | -0.023 | 0.012 | ns |
| 13 | S | CAC | 0.695 | 0.765 | 0.095 | 0.016 | 0.181 | * | 0.361 | 0.345 | -0.048 | -0.063 | -0.035 | * |
| 14 | S | CHA | 0.635 | 0.755 | 0.164 | 0.078 | 0.236 | * | 0.334 | 0.364 | 0.084 | 0.069 | 0.100 | * |
| 15 | S | TRB | 0.625 | 0.768 | 0.193 | 0.094 | 0.301 | * | 0.360 | 0.354 | -0.017 | -0.031 | -0.002 | * |
| | | Overall | 0.644 | 0.708 | 0.096 | 0.032 | 0.188 | * | 0.311 | 0.312 | 0.003 | 0.000 | 0.010 | ns |

# Estimates of within-population inbreeding based on Weir and Cockerham estimator (*f*); estimates contained in a 95% confidence interval containing zero are declared not significantly different (ns) from zero; otherwise inbreeding was declared significantly different from zero (*)

[a] Region: N = Northern population; S = Southern population

larger for microsatellites (0.644/0.708) when compared to SNPs (0.311/0.312), and the differences in value between observed and expected values for each population were much greater for microsatellites than SNPs, resulting in larger nominal estimates of inbreeding for microsatellites (Table 3). However, when the confidence intervals are taken into account, large nominal estimates of (*f*) obtained with microsatellite are not significantly different from zero for 10 out of 15 populations. On the other hand, estimates of inbreeding obtained with SNPs are considerably lower indicating very low nominal inbreeding in all populations, except CJO (*f* = 0.140) and CHA (*f* = 0.084), and not significantly different from zero in eight out of 15. Agreement between microsatellites and SNPs as far as the direction and significance of inbreeding was observed only for eight out of the 15 populations (BAR, IPI, CON, CAM, ITR, IRA, IRT and CHA). For the remaining, either disagreement as far as significance was observed (LAM, VAR, CJO, ITA and QBA), or the direction of the significant inbreeding was different (CAC and TRB). The much wider confidence intervals observed around the estimates of inbreeding with microsatellites obtained by bootstrapping over loci indicate the higher variability across loci and the much lower power of these markers to detect this occurrence. As expected, due to their larger number and the different intrinsic mutational behavior, SNPs, conversely, were able to detect significant levels of inbreeding both, positive or negative at a much finer scale. Northern populations were on average 25% less genetically diverse ($H_o$ = 0.262) than southern populations ($H_o$ = 0.353) based on SNPs, but this pattern was less pronounced with the microsatellite dataset for which the difference was only 15% (Northern $H_o$ = 0.587; Southern $H_o$ = 0.694) (Table 3).

Overall populations, a relatively high and significant inbreeding (*f* = 0.096) was estimated with microsatellites while the SNPs estimate was very low and non-significant (*f* = 0.003). As

*A. angustifolia* is outcrossed and dioecious [14] this high overall inbreeding estimated with microsatellites is definitely not expected suggesting a biologically misleading estimate. On the other hand, the trivial either positive or negative $F_{is}$ values obtained with SNPs for all populations but CJO and CHA, or the overall non-significant estimate, much better fit the expectations. The relatively high inbreeding detected with SNPs in populations CJO and CHA, possibly resulting from mating among relatives is, most likely authentic and could be explained by their distinctive topographically isolated position from the larger northern and southern Araucaria formations therefore limiting potential gene flow.

## Population differentiation: SNPs versus microsatellites

The AMOVA based partition of the genetic variation using the 2,022 SNPs indicated that ~32% of the variation is found between populations and 68% within populations, while the eight microsatellites indicated 11% between and 89% within populations (Table 4). In other words, estimates of $F_{st}$ ($F_{st} = 0.318$) were almost three times higher than with eight microsatellites ($F_{st} = 0.110$). Additionally, SNPs data estimated that the majority of the differentiation is due to the between region differentiation ($F_{stR} = 0.281$) when compared to within region between populations ($F_{stP/R} = 0.052$). Microsatellites on the other hand suggested the inverse with much closer estimates ($F_{stR} = 0.049$ and $F_{stP/R} = 0.065$). Results indicate that microsatellites underestimated population differentiation and provided an incorrect view of the relative relevance of regional versus population differentiation. To discard the potential bias due to the much larger number of SNPs to detect population differentiation, we next generated estimates from 50 random sets of 80 SNPs to estimate population parameters. The number 80 came from a general indication that around four to 12 SNPs are expected to provide the equivalent power of a single microsatellite for population structure analyses [81]. Interestingly, essentially the same average estimates of heterozygosity, inbreeding ($F_{is}$) and partitions of population differentiation were obtained with the replicates of 80 SNPs as with all 2,022 SNPs (Table 4 and S5 File). Likewise, $F_{is}$ was not significant while $F_{st}$ was highly significant with different sets of SNPs. These results show that even small numbers of SNPs are able to correctly estimate heterozigosty, inbreeding and better detect and partition the genetic variation among populations.

**Table 4. Comparative summary of genetic variation parameters and F-statistics of population differentiation via AMOVA (Analysis of Molecular Variance) for *A. angustifolia* populations using different molecular marker sets together with previously published estimates for similarly regionally located populations.**

| Marker type | # Markers | # Populations | # Regions | $H_o$ | $H_e$ | $F_{is}$ | $F_{stR}$ | $F_{stP/R}$ | $F_{stT}$ | Ref. |
|---|---|---|---|---|---|---|---|---|---|---|
| Microsatellites | 8 | 15 | 2 | 0.644 | 0.708 | 0.096 | 0.049 | 0.065 | 0.110 | this study |
| SNP (50 random sets)* | 80 | 15 | 2 | 0.316 | 0.317 | 0.003 | 0.283 | 0.050 | 0.319 | this study |
| SNP (all) | 2022 | 15 | 2 | 0.311 | 0.312 | 0.003 | 0.281 | 0.052 | 0.318 | this study |
| Isozymes | 15 | 9 | 1 | 0.073 | 0.084 | 0.148 | - | - | 0.044 | [21] |
| Isozymes | 7 | 13 | 3 | 0.132 | 0.128 | -0.036 | 0.129 | 0.013 | 0.141 | [19] |
| Microsatellites | 5 | 6 | 2 | 0.580 | 0.710 | 0.110 | - | - | 0.112 | [22] |
| AFLPs | 166 | 6 | 2 | - | 0.300 | - | - | - | 0.144 | [22] |
| Microsatellites | 15 | 12 | 2 | 0.610 | 0.670 | n.r. | - | - | 0.187# | [31] |
| cpDNA | Sequence | 39 | 3 | - | - | - | 0.521 | 0.183 | 0.700# | [82] |

$H_o$ = observed heterozygosity, $H_e$ = expected heterozygosity, $F_{is}$ = coefficient of inbreeding; $F_{stR}$ = AMOVA based $F_{st}$ between regions; $F_{stP/R}$ = AMOVA based $F_{st}$ between populations within regions; $F_{stT}$ = AMOVA based total $F_{st}$ between populations and regions

*Average estimate of 50 random sets of 80 SNP markers

# Estimate of Spatial AMOVA $F_{ct}$; n.r. not reported

To illustrate further the power of the SNP array to detect population structure analysis we compared principal coordinate analysis (PCoA) plots based on matrix of genetic distances obtained with increasing numbers of SNPs or eight microsatellites. Consistent with the much higher estimates of $F_{st}$, using all 2,022 polymorphic SNPs the 15 populations split into three distinct genetic groups (Fig 2). The axis reflecting the north-south latitudinal split is the major contributor explaining the majority of the variation (29.06%) while the additional separation of the northernmost population BAR contributes only an additional 2.07%. The microsatellite data, however, hardly detects the north-south signal explaining only 9.24% of the variation along the first PCoA axis. With just 80 randomly selected SNPs the north-south cline is clearly captured with 24.38% of the variation explained along the first axis, and population BAR already shows differentiation. With 800 SNPs, for example, the same result as using all 2,022 SNPs is already obtained (Fig 2). In a further attempt to check the possibility of separating the populations within each region using all 2,022 SNPs, individuals in the northern regions showed a clustering pattern into their respective populations, with some overlap, while in the southern regions only individuals of population 8 more clearly split from all others (S4 Fig). Our results are consistent with a review of a number of studies that compared SNPs with microsatellites showing that SNPs had greater accuracy for detecting and clustering groups of related individuals when three to several hundred fold larger numbers of SNPs are used than microsatellites [83].

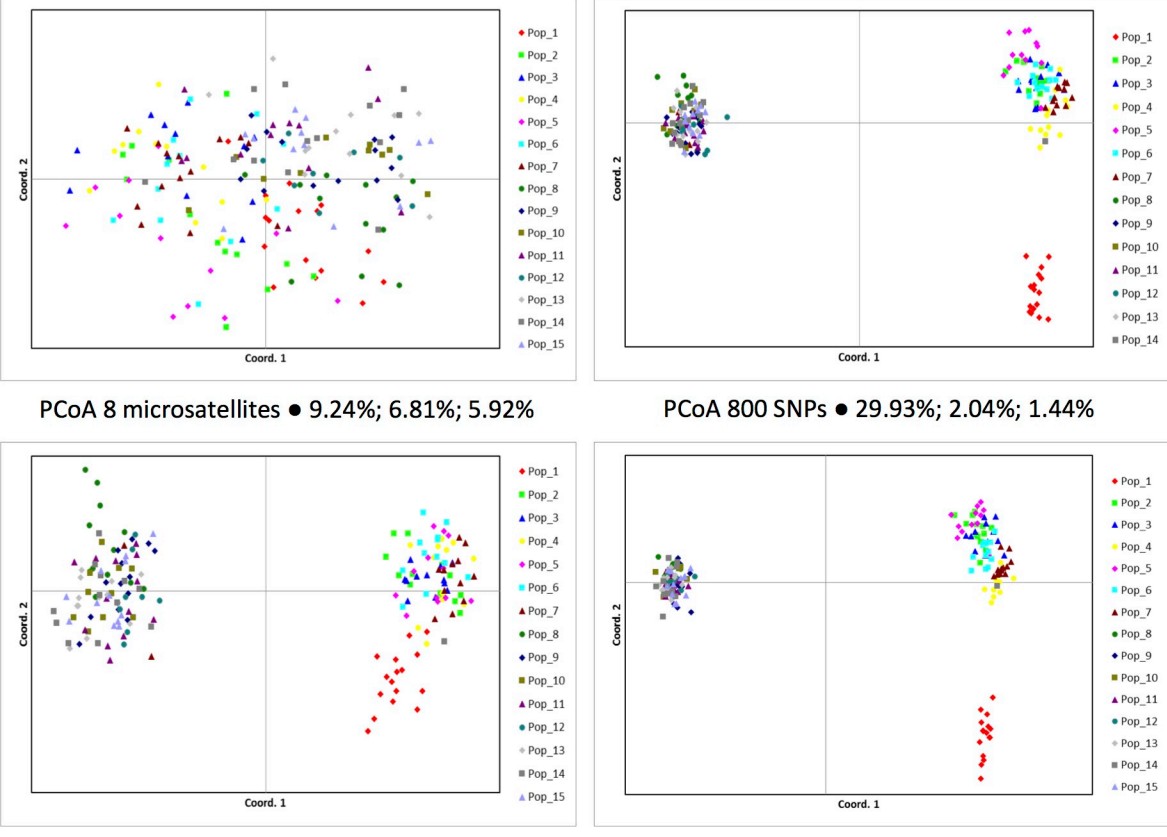

PCoA 8 microsatellites ● 9.24%; 6.81%; 5.92%

PCoA 800 SNPs ● 29.93%; 2.04%; 1.44%

PCoA 80 SNPs ● 24.38%; 3.38%; 2.65%

PCoA 2022 SNPs ● 29.06%; 2.07%; 1.39%

**Fig 2. Population structure analyses of the 15 *A. angustifolia* populations using a multidimensional principal coordinate analysis (PCoA) and different markers types (microsatellites or SNPs) and different numbers of SNPs.** The proportions of variation explained by the first three PCoA axes are indicated from left to right respectively in each plot.

Finally, we carried out an exploratory comparison of our results with previously published *A. angustifolia* studies contemplating populations with a similarly wide sampling range, both across the northern and southern regions (Table 4). The geographic distribution of remnant Araucaria populations is relatively restricted and well determined in Brazil [32], therefore suggesting valid comparisons with previous reports. Regarding population heterozygosity and inbreeding, isozymes provided considerably lower estimates than SNPs and contrasting results regarding inbreeding [19,21]. Sets of five microsatellites resulted in equally high estimates of heterozygosity with high and significant inbreeding [22,25] equivalent to what we obtained with our eight microsatellites dataset. Dominant AFLP data, on the other hand resulted in estimates of expected heterozygosity similar to SNPs while inbreeding evidently could not be estimated. Regarding population differentiation, isozymes, microsatellites and AFLPs provided equally low estimates as our microsatellite dataset. However, a recent study with 15 microsatellites showed a slightly higher estimate based on spatial AMOVA ($F_{ct}$ = 0.187) [31], suggesting that if a larger number of microsatellites is used the estimates might eventually converge to those obtained with SNPs. Chloroplast DNA, as expected, provided significantly higher estimates of population differentiation.

## STRUCTURE analyses: SNPs versus microsatellites

From the STRUCTURE analyses and based on the Evanno's 'ΔK', both the SNP and the microsatellites datasets indicated k = 2 as the most likely number of clusters, corresponding to the northern and southern groups of populations (S5 Fig). CLUMPAK generated plots of the consensus solution of the 10 independent runs are provided for all k´s tested for the different data sets (80 and 2,022 SNPs and 8 microsatellites) (S6–S8 Files). The 2,022 SNP was only able to reveal the three separate clusters at k = 3, while the microsatellites at k = 3 very clearly indicated population BAR corresponding to a third group, although at all k´s microsatellites indicated what seems an unexpected level of admixture (Fig 3). At k> 2 the 80 SNPs set also showed an unexpected level of admixture but only in the southern populations and a clear separation of population BAR at k = 4. Both SNP datasets provided a unambiguous assignment of the individuals to their respective regions, highlighting, however, one tree in the southern population CHA that most likely was mislabeled at some point, while microsatellites did not clearly detect this individual and suggested an apparent level of admixture in both clusters (Fig 3).

A review of similar microsatellite vs. SNPs comparative studies reported a number of cases of discordancy between PCoA and the number of k clusters found using different marker types and numbers [83]. In the description of the ΔK method, Evanno et al. [73] emphasized that while the method identifies the correct number of clusters in many situations, it should not be used exclusively. Furthermore, the 'ΔK' metric was found by the same authors to be sensitive to the type and number of genetic markers used, and the number of populations and individuals typed in each sample. In our study, we have strong additional evidence based on $F_{st}$ estimates and PcoA plots that three clusters exist such that this seems to be the most probable number of groups, also matching what a recent cpDNA phylogeographic study has shown [82]. Studies in humans have shown that random dinucleotide microsatellites are on average five to eight times more informative for structure and population assignment than random single-nucleotide polymorphisms (SNPs), but that a small proportion of carefully selected SNPs can be found with higher informativeness than the median for dinucleotides [81]. Considerable differences in MAF are seen between populations for several hundred SNPs indicating a good potential of selecting population specific SNP panels of ancestry informative markers (AIM) to track sample origin. Differences are seen between all pairs of populations within each region despite the overall low but significant population differentiation (p<0.01)

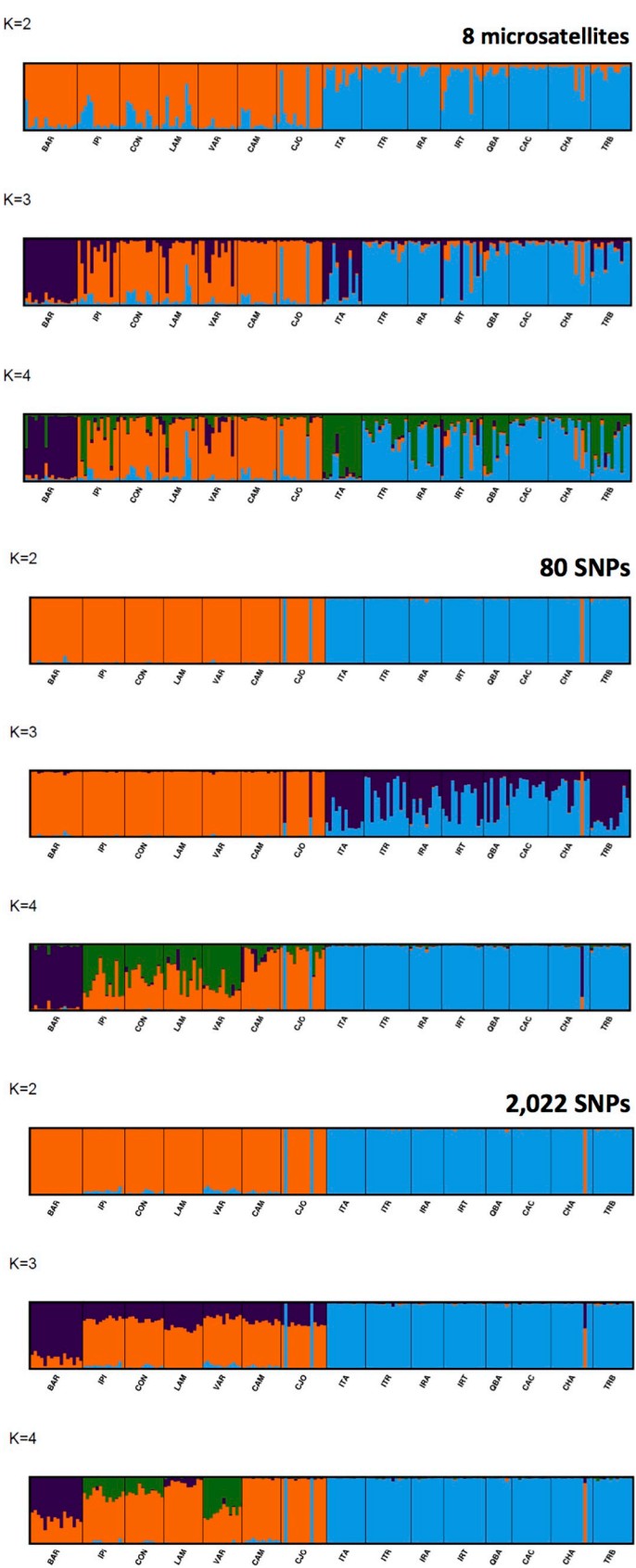

**Fig 3. Comparative population structure analyses for the 15 *A. angustifolia* populations indicated at the bottom of each panel obtained with the software STRUCTURE using a reduced (80) and full (2,022) set of SNPs and an eight-microsatellite set for different numbers of 'k' clusters (K = 2 to 4).**

as estimated by pair-wise $F_{st}$ (S2 Table). While the overall average $F_{st}$ between northern and southern populations is 0.198, between populations it is 0.057 and 0.037 within the Northern and Southern region respectively. It will be interesting in a follow up study to evaluate whether the selection of panels of AIM SNPs panels based on informative metrics of ancestry inference [81] will allow detecting additional structure and discriminating and assigning individuals to their respective populations. Panels of specific AIM SNPs on this same array could be useful, for example, to track illegal Araucaria logging by assigning timber samples to their geographic origin as shown for *Quercus* in Europe [84] or to check the origin of conserved seeds of unknown origin in germplasm banks.

## SNP array data to advance Araucaria genetics, conservation and breeding

A potential criticism of our population survey could be the relatively small samples sizes used. However, we deliberately had the goal of assessing SNP performance in a country-wide sample of as many populations possible even if smaller samples sizes had to be used to demonstrate its utility irrespective of where future studies will be carried out. While larger sample sizes have been typically used in microsatellites surveys, with SNP genotyping this might not be strictly necessary, adding a further advantage to the use of large SNP sets. A number of studies have shown that large numbers of biallelic SNPs ($>$ 1000) efficiently compensate for small sample sizes as small as n = 2 to 6 [85–88]. We are therefore confident that our results of population diversity and differentiation with 2,022 SNPs are robust and comparable with previous studies.

The SNP data gathered validate the ample applicability of the Axiom array and casts new light on the overall picture of diversity and structure of *A. angustifolia* along its extensive natural range. SNPs more accurately showed that southern populations ($H_o$ = 0.353) are significantly more genetically diverse that northern populations ($H_o$ = 0.262) (Table 3), consistent with recent indications [31]. When compared to microsatellites, SNPs provided considerably lower estimates of heterozigosity and scant, if any, evidence of trivial inbreeding either positive or negative within-populations, and none at range wide level. While microsatellites have frequently delivered large nominal estimates of $F_{is}$, SNPs provide precise estimates that allow confidently detecting even minor levels of inbreeding if they in fact exist (Table 3). The major north-south genetic cline detected by SNPs (Figs 2 and 3) is consistent with previous reports based on isozymes [19] and microsatellites [22]. Nevertheless, SNPs discriminated these two groups with a considerably higher magnitude (Table 4), together with the identification of a third PCoA cluster (Fig 2) and further confirmed by a STRUCTURE analysis, robustly assigning individuals to their respective regions (Fig 3), and showing a considerably higher population differentiation due to regional than population within-region difference (Table 4). This genetic cline has been explained by the combined effect of post-glacial migration from different refugia [22,82] and a north-south isolation most likely due to niche suitability [31]. The possibility of an additional phylogeographic separation of a third genetic group in the northern populations was also suggested earlier based on tenuous evidence from microsatellites [22] and recently confirmed based on sequence data of three intragenic regions in the chloroplast genome (cpDNA) [82]. Interestingly, the 2,022 autosomal SNP data set was able to provide equally strong evidence as the chloroplast sequence-based data for a third group, therefore matching the stronger phylogenetic signal typically obtained from the uniparental non-recombining inheritance of cpDNA [89].

The somewhat conflicting results between SNPs and microsatellite based population esti-mates are not surprising and should not be taken as criticisms to previous studies in *Araucaria angustifolia*. The four to six orders of magnitude higher mutation rate of microsatellites when compared to SNPs [35], and the consequential allele hypervariability will tend to result in higher heterozygosities. Moreover, the twice higher estimates of genetic diversity in *A. angusti-folia* obtained with microsatellites when compared to SNPs (Table 4) may also be a conse-quence of a strong ascertainment bias when selecting the most polymorphic markers [78]. Combined with the small number of markers that sample a narrow fraction of the genome, biased estimates of genetic diversity may result [33]. A microsatellite-SNP comparative analy-sis in the outcrossed *Arabidopsis halleri* for example, showed the same pattern of twice to three times higher estimates of heterozigosity with microsatellites when compared to SNPs. More importantly, however, was the fact that while the microsatellite heterozigosity showed no cor-relation at all with the canonical Watterson theta (θ) genetic diversity metric, the SNP-based heterozigosity was highly correlated, properly reflecting the genome-wide genetic diversity [90]. Concerning the considerably higher estimates of $F_{st}$ obtained with SNPs when compared to microsatellites, the recurrent mutation of the latter frequently leads to homoplasic alleles which are identical by state (size) but not identical by descent [91]. This fact tends to dampen the signal of population structure that SNPs correctly capture, even when only small random subsets of only 80 out of the 2,022 SNPs were used (Fig 2 and S5 File). A number of previous studies in plant and animal species have in fact shown higher $F_{st}$ estimates with SNPs when compared to microsatellites and a superior ability of SNPs to resolve fine-scale population structure and phylogeographic signals [88,92,93].

## Conclusions

To summarize, we have developed the first comprehensive SNP resource for *Araucaria angu-stifolia*, a keystone subtropical conifer tree, critically endangered due to its valuable wood and seeds. From the transcriptome-wide catalog of 44,318 annotated SNP an Axiom® SNP array with ~3,000 validated SNPs was developed as part of a multi-species SNP array strategy signifi-cantly reducing the individual sample genotyping cost therefore allowing access to a high qual-ity SNP array even for this generally underfunded species. Data obtained with this newly developed SNP genotyping platform provided a comprehensive look at the range-wide genetic diversity and structure of the species. By matching SNP with microsatellite data on the exact same individuals, our results indicate that microsatellite markers may have led to estimates of genetic diversity and differentiation that have not precisely reflected the actual genome-wide patterns of variation and structuring. The generally established microsatellite-based inference that *A. angustifolia* has been resilient against rapid losses of its genetic diversity due to forest fragmentation might not be fully warranted and should receive further attention [30,32]. Over-estimated diversity, inaccurate inbreeding estimates and underestimated population differenti-ation can have relevant consequences on decisions on how to approach and manage *ex situ* and *in situ* conservation of the species' genetic resources. Our results do not doubt the useful-ness of microsatellites in general as efficient tool for studying mating systems, kinship and relatedness at low taxonomic levels, but we caution on taking their diversity, inbreeding and differentiation estimates at face value given their known limitations. Because genetics applica-tions rely on multilocus estimators of differentiation, panels of several hundred to thousand genome-wide SNPs will always be more powerful, representative and accurate than a dozen microsatellites [1]. It is relevant to note, however, that even SNPs when genotyped using meth-ods based on sequencing reduced genomic representations, face significant challenges to pro-vide consistent and interchangeable SNP genotypes, limiting, for example, valid across-study

comparisons and meta-analyses due to the stochastic genome sampling and the sources of sequencing bias involved in these techniques [94].

The availability of a public, user-friendly 3K SNP array for *A. angustifolia* and the catalog of 44,318 SNPs predicted to provide ~29,000 informative SNPs across ~20,000 loci across the genome, will allow tackling still unsettled questions on its evolutionary history, toward a more comprehensive picture of the origin, past dynamics and future trend of the species' genetic resources. Additionally, but not less importantly, the SNP array described, unlocks the potential to consider adopting genomic prediction methods [95] to accelerate the still very timid efforts of systematic tree breeding of *A. angustifolia*, a species with enormous potential for its valuable wood and seed products but with very long generation times. In conclusion, this first fully public fixed content SNP array for *A. angustifolia* and the additional extensive SNP catalog provided in this work for the future manufacture of even denser arrays, raises this iconic species to a higher level for genetic research, opening opportunities to increase the breadth, precision, long-term portability and impact of the genetic data generated.

## Supporting information

**S1 File. Catalog of all 44,318 high-quality SNPs detected from the RNA-seq and RAD-seq data with their corresponding probes designed and annotated contig.**
(XLSX)

**S2 File. List of the 3,400 SNPs on the Axiom array with their corresponding SNP probe and performance data (Call Rate, MAF, Axiom performance criteria).**
(XLSX)

**S3 File. Data of the 2,022 SNPs used in the analyses with estimates of allele frequencies, observed (Ho) and expected (He) heterozygosity and test for hardy weinberg equilibrium.**
(XLSX)

**S4 File. Data of the 8 microsatellites used in the analyses and estimates of allele frequencies, observed (Ho) and expected (He) heterozygosity and test for hardy weinberg equilibrium.**
(XLSX)

**S5 File. Estimates of heterozygosity ($H_o$ e $H_e$) within-population inbreeding ($Fi_s$), fixation index ($F_{st}$) and total reduction of heterozigosity ($F_{it}$) obtained with the 50 replicates of 80 randomly selected SNPs.**
(XLSX)

**S6 File. CLUMPAK generated plots of the consensus solution of the 10 independent runs for all 15 k´s tested using 8 microsatellites.**
(PDF)

**S7 File. CLUMPAK generated plots of the consensus solution of the 10 independent runs for all 15 k´s tested using 80 SNPs.**
(PDF)

**S8 File. CLUMPAK generated plots of the consensus solution of the 10 independent runs for all 15 k´s tested using 2,022 SNPs.**
(PDF)

**S1 Table. Description of the genotyped samples from different populations of *Araucaria angustifolia* used to validate the 3K SNP Axiom®️ Array.** City and State of origin, number of samples genotyped, latitude, longitude and altitude of the populations sampled are

provided.
(DOC)

**S2 Table. Heat map of pairwise $F_{st}$ estimates based on 2,022 polymorphic SNPs among all 15 populations indicating the higher differentiation between the populations in the northern (1 to 7) and southern (8 to 15) regions, and lower differentiation between populations within regions.** Populations 7 and 8 in the transition zone display slightly differentiated $F_{st}$ estimates from their regionally associated populations as indicated by the heatmap. All $F_{st}$ estimates were significant (p<0.001) based on a permutation test by bootstrapping over loci.
(DOC)

**S1 Fig. Top 20 most abundant InterPro domains and families identified in the *A. angustifolia* non-redundant gene set.** (A) InterPro domain annotation was performed on the non-redundant gene set obtained from the transcriptome assembly of *A. angustifolia*. (B) InterPro family annotation was performed on the non-redundant gene set obtained from the transcriptome assembly of *A. araucaria*.
(DOC)

**S2 Fig. Distribution of most abundant gene ontology (GO) terms in the three GO categories assigned to the *Araucaria angustifolia* contigs.** Only level 3 terms are represented.
(DOC)

**S3 Fig. SNP frequency spectrum of all 3,038 successfully genotyped SNPs by the two performance evaluation criteria.** Also plotted are only the SNPs derived from RNA-seq data to show that the majority of monomorphic SNPs came from RAD-seq data.
(DOC)

**S4 Fig. Population structure analyses using a multidimensional Principal Coordinate analysis (PCoA) with the full set of 2,022 SNPs.** Top panel: PCoA plot involving only the seven northern region populations; Bottom panel: PCoA plot involving only the eight southern region populations. The proportions of variation explained by the first three PCoA axes are indicated from left to right respectively in each plot.
(DOC)

**S5 Fig. Results of Evanno´s Delta K analysis to define the most probable number of populations with different sets of markers as indicated in the figure.**
(DOC)

## Acknowledgments

We acknowledge the continued suport for research provided by FAP-DF (Foundation for Scientific Research of the Federal District) and CNPq (Brazilian National Council for Scientific and Technological Development). We would like to thank Ananias de Almeida S. Pontinha, Miguel L. Menezes Freitas and the field staff of the Instituto Florestal de São Paulo (Itapeva experimental station) for technical and logistic support for field sample collection.

## Author Contributions

**Conceptualization:** Dario Grattapaglia.

**Data curation:** Pedro Italo T. Silva, Orzenil B. Silva-Junior, Lucileide V. Resende, Valderes A. Sousa, Ananda V. Aguiar.

**Formal analysis:** Pedro Italo T. Silva, Orzenil B. Silva-Junior, Lucileide V. Resende, Dario Grattapaglia.

**Funding acquisition:** Valderes A. Sousa, Ananda V. Aguiar, Dario Grattapaglia.

**Investigation:** Pedro Italo T. Silva, Lucileide V. Resende.

**Methodology:** Pedro Italo T. Silva, Orzenil B. Silva-Junior, Lucileide V. Resende, Ananda V. Aguiar.

**Project administration:** Valderes A. Sousa, Dario Grattapaglia.

**Resources:** Valderes A. Sousa, Ananda V. Aguiar, Dario Grattapaglia.

**Software:** Orzenil B. Silva-Junior.

**Supervision:** Dario Grattapaglia.

**Validation:** Pedro Italo T. Silva, Orzenil B. Silva-Junior, Lucileide V. Resende, Ananda V. Aguiar.

**Writing – original draft:** Pedro Italo T. Silva.

**Writing – review & editing:** Orzenil B. Silva-Junior, Valderes A. Sousa, Ananda V. Aguiar, Dario Grattapaglia.

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
