## [Decision Letter · Decision Letter 0]

2 Apr 2020

PONE-D-20-05745

A 3K Axiom® SNP array from a transcriptome-wide SNP resource sheds new light on the genetic diversity and structure of the iconic subtropical conifer tree Araucaria angustifolia (Bert.) Kuntze

PLOS ONE

Dear Dr. Grattapaglia,

Thank you for submitting your manuscript to PLOS ONE. After careful consideration, we feel that it has merit but does not fully meet PLOS ONE’s publication criteria as it currently stands. Therefore, we invite you to submit a revised version of the manuscript that addresses the points raised during the review process.

We would appreciate receiving your revised manuscript by May 17 2020 11:59PM. To enhance the reproducibility of your results, we recommend that if applicable you deposit your laboratory protocols in protocols.io, where a protocol can be assigned its own identifier (DOI) such that it can be cited independently in the future. For instructions see: http://journals.plos.org/plosone/s/submission-guidelines#loc-laboratory-protocols

We look forward to receiving your revised manuscript.

Kind regards,

Peng Xu, Ph.D.

Academic Editor

PLOS ONE

2. We understand that samples were taken from a species listed as Critically Endangered by the IUCN. In your Methods section, please provide additional information regarding the permits you obtained for the work. Please ensure you have included the full name of the authority that approved the field site access and, if no permits were required, a brief statement explaining why.

Reviewers' comments:

Reviewer's Responses to Questions

**Comments to the Author**

1. Is the manuscript technically sound, and do the data support the conclusions?

Reviewer #1: Yes

2. Has the statistical analysis been performed appropriately and rigorously? 

Reviewer #1: Yes

3. Have the authors made all data underlying the findings in their manuscript fully available?

Reviewer #1: Yes

4. Is the manuscript presented in an intelligible fashion and written in standard English?

Reviewer #1: No

5. Review Comments to the Author

Reviewer #1: For the writing, a lot of very long and complex sentences were included in this manuscript, making it difficult to read and understand. I suggest the authors to revise them and use more short sentences instead.

6. PLOS authors have the option to publish the peer review history of their article (what does this mean?). If published, this will include your full peer review and any attached files.

Reviewer #1: No

---

## [Author Response · Author response to Decision Letter 0]

4 May 2020

RESPONSE TO EDITORIAL AND REVIEWERS REQUIREMENTS - PONE-D-20-05745

“A 3K Axiom® SNP array from a transcriptome-wide SNP resource sheds new light on the genetic diversity and structure of the iconic subtropical conifer tree Araucaria angustifolia (Bert.) Kuntze”

EDITORIAL REQUIREMENT

EDITOR: We understand that samples were taken from a species listed as Critically Endangered by the IUCN. In your Methods section, please provide additional information regarding the permits you obtained for the work. Please ensure you have included the full name of the authority that approved the field site access and, if no permits were required, a brief statement explaining why.

RESPONSE: Following the editorial requirements, we have included in the Methods section the full information regarding the governmental authorization we received to develop this research with Araucaria angustifolia being it a critically endangered species by the IUCN. The authorization number 02001.007609/2012-77 was issued by the Brazilian Institute of the Environment (IBAMA), the regulating body of the Brazilian Ministry of Environment in December 2012 giving full authorization to collect samples from natural populations for DNA analysis toward genetic studies

REVIEWER #1

REVIEWER #1: In the SNP filtering step, Line 221, the parameters of minimum allele frequency (MAF), missing rate, and Hardy-Weinberg equilibrium (HWE) were not used. The MAF filtering can exclude SNPs showing little variation in the population. The SNPs with a high missing rate indicated the number of detected genotype in the population is limited. The HWE filtering help exclude the SNPs arising from sequencing errors and natural selection. Therefore, these criteria are important for the selection of the SNPs. 

RESPONSE: The section that the reviewer is indicating on line 221 does not refer to the filtering of SNPs following genotyping. Rather it describes the steps taken during the SNP discovery process using the VCF files of the RAD and RNA sequencing data or a few individuals used for that objective. The parameters taken during SNP discovery are evidently different ones based sequence context and parameters implemented by the GATK best practices. Also given that only a few individuals were sequenced for SNP discovery, neither MAF nor HWE would make any sense.

The parameters the reviewer is mentioning (MAF, call rate and HWE) were used after SNP genotyping is carried out with the array on population samples. This was done and fully explained in section “ SNP array design and validation” on line 267. In that section these parameters were applied during the SNP validation process that followed the construction of the SNP array. 

Still regarding the HWE filter, although this filter makes sense in human genetics or species that abide to the HWE expectations given a dioecious sexual reproduction, large populations and little or no structure, this filter does not apply to plant populations with mixed mating or selfing system in subdivided populations. For example, in selfing species, if a HWE filter is applied, all SNPs would be excluded as no or very few heterozygotes are seen. Additionally, in our case, since we genotyped individuals from several different populations, the consolidated set of data would never abide to HWE due to strong population differentiation that causes reduction n overall heterozygosity. So the HWE filter cannot be applied in our case as in essentially all plant population SNP arrays.

REVIEWER #1: In Line 285, 294, 408, MAF ≥ 0.005 was used to declare a polymorphic SNP. However, the commonly recognized and used MAF threshold is 0.05 or 0.01 for valid SNPs. MAF ≥ 0.005 indicated that the array may contain a number of rare SNPs. A MAF distribution plot should be provided to help evaluate this issue.

RESPONSE: We do acknowledge that commonly the MAF threshold used is MAF> 0.05. However this is really a threshold that depends on the sample size used and the envisaged applications of the SNP array. As we describe in detail in that same section “ SNP array design and validation” we used a call rate ≥ 97% or ≥ 90% depending on the criteria adopted and MAF ≥ 0.005 for two reasons. We did deliberately intend to include rare SNPs in our array as these are in fact informative especially if one intends to use the SNP array for association analysis. 

Low-frequency and rare variants have been shown to represent an important and understudied component of complex trait genetics. The literature on the importance of rare variants is abundant especially in human genetics (see for example doi: 10.1126/science.1219240 and doi: 10.1007/978-1-4939-7868-7_5) and methods have been increasingly been developed and used to deal with rare variants in GWAS (see for example doi: 10.1002/cphg.83). The second reason is that we genotyped different Araucaria populations and SNP allele frequencies vary across populations such that a SNP may be rare in one population and frequent in another. This is commonly seen in plant populations across large geographical areas mainly due to drift and occasionally selection. So the information content of SNPs will vary across populations and this fully justifies adopting a much lower MAF threshold to include all SNPs with such behavior. The genotyping quality of the SNPs was evaluated by the dish quality parameter that has to do with the quality of cluster separation among other things. 

Regarding a MAF distribution plot, which we called site frequency spectrum (SFS) plot, it was adequately provided this in our supplementary material, Figure S3 and cited in the manuscript on line 427. As can be seen in the figure, the SFS does in fact show an enrichment on rare variants. As explained above, this was fully intentional and justified. However the plot also shows enrichment toward higher frequency SNPs.

REVIEWER #1: In Line 271, the author mentioned that the 3K SNP probes for the conifer were tilled on a multispecies Axiom® myDesign ™ array that contained a total of 51,867 SNPs, shared among five different plant species. What is the phylogenetic relationship of these species and more importantly, the genetic divergence of these species? If they are closely related species, some SNPs may inter-influence and cause false genotyping result. 

RESPONSE: We would like to thank the reviewer for this question, which provides an opportunity to explain what we did. We added a sentence in Methods explaining this issue, to better clarify the readership.

The development of this multispecies array will be the topic of a specific methods publication under preparation. For right now we presented it as an invited talk at Plant Genome which can be viewed in Youtube (https://www.youtube.com/watch?v=-Wfb6WftQiU), and the reference was cited in the manuscript (reference 62). The five species were Araucaria angustifolia (Brazilian pine – order Pinales), Anacardium occidentale (cashew – order Sapindales), Manihot esculenta (cassava – order Malpighiales), Coffea robusta (coffee – order Gentianales) and Eucalyptus sp. (order Myrtales). These species are phylogenetically quite distant. Not only they belong to different families but also to different phylogenetic orders. Additionally, during array design all 51 thousand plus probes designed were subject to a detailed bioinformatics analysis to avoid probe homology such that SNP probe cross talking was precluded. Finally, even in the unlikely case that some cross talk signal happens, an analytical step can be used in Axiom Suite to quench the fluorescent signal of unwanted probes such that only the signal of the probes of the species under analysis is captured. Reminding, evidently, that only a single species sample is hybridized to the array at the time, in other words no DNA mixing is used. So we are fully confident that no false genotyping result happened or will happen for any of the five species.

---

## [Decision Letter · Decision Letter 1]

24 Jul 2020

PONE-D-20-05745R1

A 3K Axiom® SNP array from a transcriptome-wide SNP resource sheds new light on the genetic diversity and structure of the iconic subtropical conifer tree Araucaria angustifolia (Bert.) Kuntze

PLOS ONE

Dear Dr. Grattapaglia,

Thank you for submitting your manuscript to PLOS ONE. After careful consideration, we feel that it has merit but does not fully meet PLOS ONE’s publication criteria as it currently stands. Therefore, we invite you to submit a revised version of the manuscript that addresses the points raised during the review process.

We look forward to receiving your revised manuscript.

Kind regards,

Xiaoming Pang, PhD

Academic Editor

PLOS ONE

Reviewers' comments:

Reviewer's Responses to Questions

**Comments to the Author**

1. If the authors have adequately addressed your comments raised in a previous round of review and you feel that this manuscript is now acceptable for publication, you may indicate that here to bypass the “Comments to the Author” section, enter your conflict of interest statement in the “Confidential to Editor” section, and submit your "Accept" recommendation.

Reviewer #1: All comments have been addressed

Reviewer #2: (No Response)

2. Is the manuscript technically sound, and do the data support the conclusions?

Reviewer #1: Yes

Reviewer #2: Yes

3. Has the statistical analysis been performed appropriately and rigorously? 

Reviewer #1: Yes

Reviewer #2: Yes

4. Have the authors made all data underlying the findings in their manuscript fully available?

Reviewer #1: Yes

Reviewer #2: Yes

5. Is the manuscript presented in an intelligible fashion and written in standard English?

Reviewer #1: Yes

Reviewer #2: Yes

6. Review Comments to the Author

Reviewer #1: For the minor allele frequency, the authors have raised two reasons for the selction of 0.005. However, since the total number of the SNPs tilling on the array is only 3k, the rare SNPs are difficult to represent significance in both GWAS and in different populations. I still think a higher filtering parameter for MAD is necessary to ensure a higher genotyping rate.

Reviewer #2: Because the huge genome of conifers, cheap array based genotyping rather than large scale GBS will serve as an excellent foundation for implementing genomic selection. Therefore, this article provides useful information to researchers targeting at conifer species. I think that this article should be published.

Minor comments

1. Line 60, “ low to ultra-low sequencing of the whole genome.” what is low sequencing? Do you mean “low-coverage” or “low-cost”?

2. line 78, I disagree with this comment. I think that in addition to humans and animals, many forest trees already have high-density SNP chips for years, including conifers (Mol Ecol Resour. 2013;13(2):324–36. BMC Genomics, 2020;21:9)

3. line 168, the StringTie was used for transcriptome assembly, I think this tool need BAM format data that aligned reads to genome as input, so any references genome was available in this study?

4. It is true that SNP will definitely outperform SSR because of its large number, but comparing thousands of SNPs with just 8 SSR to make this point is neither fair nor necessary.

5. line 486, “Northern populations were on average 25% less genetically diverse (Ho = 0.262) than southern populations (Ho= 0.353) based on SNPs, but this pattern was not as clearly detected with the microsatellite dataset” , I am not sure if this conclusion is correct. Based on statistical tests, I guess they all have significant differences.

7. PLOS authors have the option to publish the peer review history of their article (what does this mean?). If published, this will include your full peer review and any attached files.

Reviewer #1: No

Reviewer #2: No

---

## [Author Response · Author response to Decision Letter 1]

29 Jul 2020

RESPONSE TO ADDITIONAL REVIEWERS REQUIREMENTS - PONE-D-20-05745R1

“A 3K Axiom® SNP array from a transcriptome-wide SNP resource sheds new light on the genetic diversity and structure of the iconic subtropical conifer tree Araucaria angustifolia (Bert.) Kuntze”

REVIEWER #1

REVIEWER #1 COMMENT: For the minor allele frequency, the authors have raised two reasons for the selection of 0.005. However, since the total number of the SNPs tilling on the array is only 3k, the rare SNPs are difficult to represent significance in both GWAS and in different populations. I still think a higher filtering parameter for MAF is necessary to ensure a higher genotyping rate.

RESPONSE: As we had responded previously, the threshold used was in fact justified by the two reasons we outlined before. On one hand we deliberately intended to include rare SNPs in our array as these could be informative in specific situations. However the strongest justification was that the estimates were obtained in limited samples sizes for each one of several populations, and in light of the strong differentiation detected, it is likely that SNP allele frequencies vary across populations, such that a SNP allele may be rare overall but it may have higher frequency in some specific populations and thus informative. So, these rare alleles could in fact be well represented in some populations when larger samples sizes are genotyped. This is commonly seen in outbred plant populations across large geographical areas mainly due to drift and occasionally selection.

Regarding the second point we do agree with the reviewer that to include rarer SNPs it is important to ensure high genotyping rate. As described in the MS, we did use stringent dish quality and call rates metrics supported by the Axiom technology (Call rate > 97% for Axiom criteria) such that the quality of the genotype call is robust, despite the low estimated MAF. So we really don’t see any reason to exclude these seemingly rare SNPs from our evaluation in this initial array performance. Certainly as larger samples are studied across variable populations we are confident that several of these rarer SNPs will display higher frequencies.

Another point that should be mentioned is that by including rarer SNPs on the array, the ascertainment bias toward having largely frequent SNPs typical seen in array design, gets mitigated somehow adding to the value of the array. We had discussed this point in the MS.

In any case to address the reviewer’s concern, we added a paragraph to the MS, based on our arguments above, further detailing the reasons why we used a lower MAF threshold. We are confident that this would clarify the readership about our decision. We wrote:

“Despite the fact that typically only SNPs with MAF ≥ 0.05 are considered as polymorphic, we deliberately intended to take into account rare SNPs in our array for two reasons: (1) rare SNPs could be informative for some specific applications such as gene flow, parentage and association studies; (2) we used stringent dish quality and call rates metrics criteria such that the quality of the genotype calls are robust, despite the low MAF; and (3) by having genotyped limited samples of individuals in this first study for each one of several populations, and in light of the strong differentiation detected (see below), it is expected that SNP allele frequencies will vary across populations, such that a SNP allele may be rare overall but it may have higher frequency and thus be informative in specific populations. So, these rare SNP alleles could turn out to be well represented in some populations when larger samples sizes are genotyped, especially for outbred forest tree populations across large geographical areas.”

REVIEWER #2

Minor comments

REVIEWER #2 COMMENT 1. Line 60, “ low to ultra-low sequencing of the whole genome.” what is low sequencing? Do you mean “low-coverage” or “low-cost”?

RESPONSE: Thanks. We really mentioned low-coverage sequencing which also tend to be low-cost from a general point of view. We fixed that.

REVIEWER #2 COMMENT 2. line 78, I disagree with this comment. I think that in addition to humans and animals, many forest trees already have high-density SNP chips for years, including conifers (Mol Ecol Resour. 2013;13(2):324–36. BMC Genomics, 2020;21:9)

RESPONSE: Yes we agree and we added forest trees to the sentence. Additionally, as can be read a few paragraphs down the introduction, we had fully cited all the conifer chips published up to the time we had submitted the MS, including the Mol Ecol Res 2013 reference mentioned. We have now added the Howe et al. 2020 reference to the citations to make it fully complete.

REVIEWER #2 COMMENT 3. line 168, the StringTie was used for transcriptome assembly, I think this tool need BAM format data that aligned reads to genome as input, so any references genome was available in this study?

RESPONSE: Thank you for this opportunity to better clarify our methods in the manuscript. In fact, we have not used StringTie for the transcriptome assembly. As described in the prior MS version, we have adopted the StringTie approach only as an attempt to reconstruct cDNA fragments for both ends of the reads from the RNA-Seq data. The reviewer is correct to notice that StringTie uses BAM formatted data as input and also requires reads to be aligned to a reference genome. However, the StringTie tool includes a separate routine called SR (from Super-Reads), in which the input is constituted of unaligned paired-end reads. This is an optional step for preparing the alignments for the input of StringTie+SR runs (described in the StringTie manuscript, (https://www.ncbi.nlm.nih.gov/pmc/articles/PMC4643835/).

StringTie+SR borrows some algorithmic techniques from de novo genome assembly (i.e, do not need an available reference genome) to help with transcript assembly. Its input can include longer contigs that it assembles de novo from unambiguous, non-branching parts of a transcript. If the RNA-Seq data is paired, it is possible to use the SR routine to reconstruct the RNA-Seq fragments from their end sequences, which the authors of the tool have called Super-Reads (please see the StringTie+SR manual, 

http://ccb.jhu.edu/software/stringtie/index.shtml?t=manual#assembly). 

It can be used as an optional step for preparing these longer contigs for the input of StringTie+SR or any other transcriptome assembly tool that can handle single-end longer contigs that correspond approximately to the original cDNA fragment (such as an EST assembler tool). Because the StringTie+SR still relies on the alignment of these Super-Reads on the top of an reference genome, which is not available in the case of the Araucaria angustifolia, we just use this data as an input of the miraSearchESTSNPs assembler as described later in the text. To further clarify these points, we rewrote this section in the manuscript as follows: 

“As a first step, we used the StringTie+SR [51] method to pre-assemble longer contigs from the RNA-Seq paired-end data aiming to achieve the better performance of the subsequent transcriptome assembly. From StringTie+SR, we ran the perl script SR (Link to external file: http://ccb.jhu.edu/software/stringtie/dl/superreads.pl) separately on the reads of each three megastrobiles and each three somatic embryogenic cultures of A. angustifolia. The SR method pre-assembled de novo the paired-end reads from the unambiguous, non-branching parts of the transcript in these six sources of cDNA data, resulting in super-reads of approximately 300 bp corresponding to the entire sequence of the original cDNA fragment. These super-reads were used as input of the miraSearchESTSNPs from MIRA 4 assembler [52] to generate the consensus sequences of the transcript assembly from the different sources of cDNA and to perform the SNP detection within this assembly.”

REVIEWER #2 COMMENT 4. It is true that SNP will definitely outperform SSR because of its large number, but comparing thousands of SNPs with just 8 SSR to make this point is neither fair nor necessary.

RESPONSE: We definitely agree that 8 microsatellites are not comparable to the power of a large set of SNPs. However the point we intend to make in this study is valid in light of what has been published in the literature so far regarding the population genetics structure and diversity of A. angustifolia. Essentially all studies to date have used small sets of 5 to 10 microsatellites and this is the reason we made this comparison. This is in fact a crucial point of our MS, besides the SNP array development. It is therefore necessary, in our understanding, to clearly compare results obtained with the new array we present with was has been done so far to make the point that several of the conclusions provided in the literature for the population genetics of A. angustifolia are, in fact, misrepresentative. 

To this end we made sure we genotyped the same exact individuals with the two methods so that no doubt would persist to the reader regarding population and individual sampling and the conclusions derived thereof regarding diversity, inbreeding and structure of the species. So we thank the reviewer for the comment but we do think it is necessary to keep this analysis exactly to make a strong point about this issue. We go over this in detail in our discussion especially in the final section “SNP array data to advance Araucaria genetics, conservation and breeding”. We discuss that the consequences of deriving conclusions about diversity and structure from microsatellite data could be deceptive to any recommendation for conservation.

REVIEWER #2 COMMENT 5. line 486, “Northern populations were on average 25% less genetically diverse (Ho = 0.262) than southern populations (Ho= 0.353) based on SNPs, but this pattern was not as clearly detected with the microsatellite dataset” , I am not sure if this conclusion is correct. Based on statistical tests, I guess they all have significant differences.

RESPONSE: Thanks for the comment and observation. We agree that the difference also existed but was quite smaller. We improved this sentence in the revised version by stating: “Northern populations were on average 25% less genetically diverse (Ho = 0.262) than southern populations (Ho= 0.353) based on SNPs, but this pattern was less pronounced with the microsatellite dataset for which the difference was only 15% (Northern Ho = 0.587; Southern Ho = 0.694) (Table 3).”

---

## [Editor Report · Decision Letter 2]

6 Aug 2020

A 3K Axiom® SNP array from a transcriptome-wide SNP resource sheds new light on the genetic diversity and structure of the iconic subtropical conifer tree Araucaria angustifolia (Bert.) Kuntze

PONE-D-20-05745R2

Dear Dr. Grattapaglia,

We’re pleased to inform you that your manuscript has been judged scientifically suitable for publication and will be formally accepted for publication once it meets all outstanding technical requirements.

Kind regards,

Xiaoming Pang, PhD

Academic Editor

PLOS ONE
---

## [Editor Report · Acceptance letter]

12 Aug 2020

PONE-D-20-05745R2 

A 3K Axiom® SNP array from a transcriptome-wide SNP resource sheds new light on the genetic diversity and structure of the iconic subtropical conifer tree *Araucaria angustifolia* (Bert.) Kuntze 

Dear Dr. Grattapaglia:

I'm pleased to inform you that your manuscript has been deemed suitable for publication in PLOS ONE. Congratulations! Your manuscript is now with our production department. 

Kind regards, 

on behalf of

Dr. Xiaoming Pang 

Academic Editor

PLOS ONE